# Characteristics and Outcomes of Patients Consulted by a Multidisciplinary Pulmonary Embolism Response Team: 5-Year Experience

**DOI:** 10.3390/jcm11133812

**Published:** 2022-06-30

**Authors:** Arkadiusz Pietrasik, Aleksandra Gąsecka, Paweł Kurzyna, Katarzyna Wrona, Szymon Darocha, Marta Banaszkiewicz, Dariusz Zieliński, Dominika Zajkowska, Julia Maria Smyk, Dominika Rymaszewska, Karolina Jasińska, Marcin Wasilewski, Rafał Wolański, Grzegorz Procyk, Piotr Szwed, Michał Florczyk, Krzysztof Wróbel, Marcin Grabowski, Adam Torbicki, Marcin Kurzyna

**Affiliations:** 11st Chair and Department of Cardiology, Medical University of Warsaw, 02-097 Warsaw, Poland; gaseckaa@gmail.com (A.G.); paw.kurzyna@gmail.com (P.K.); dominika_zajkowska@interia.pl (D.Z.); julia.lekarski@gmail.com (J.M.S.); dmnrymaszewska@gmail.com (D.R.); kari.jasinska@gmail.com (K.J.); marcin.vasilewski@wp.pl (M.W.); rafalwolanski7@gmail.com (R.W.); grzegorzprocyk@gmail.com (G.P.); szwedp12@gmail.com (P.S.); marcin.grabowski@wum.edu.pl (M.G.); 2Department of Pulmonary Circulation, Thromboembolic Diseases and Cardiology, Centre of Postgraduate Medical Education, European Health Centre Otwock, 05-400 Otwock, Poland; kasia.wrona18@gmail.com (K.W.); szymon.darocha@gmail.com (S.D.); marta.banaszkiewicz@gmail.com (M.B.); michal.florczyk@ecz-otwock.pl (M.F.); adam.torbicki@ecz-otwock.pl (A.T.); marcin.kurzyna@ecz-otwock.pl (M.K.); 3Department of Cardiac Surgery, Medicover Hospital, 02-972 Warsaw, Poland; farok@wp.pl (D.Z.); krzysztof.wrobel17@gmail.com (K.W.)

**Keywords:** pulmonary embolism, pulmonary embolism response team, PERT, catheter-based therapies

## Abstract

(1) Background: Pulmonary embolism (PE) is the third most frequent acute cardiovascular condition worldwide. PE response teams (PERTs) have been created to facilitate treatment implementation in PE patients. Here, we report on the 5-year experience of PERT operating in Warsaw, Poland, with regard to the characteristics and outcomes of the consulted patients. (2) Methods: Patients diagnosed with PE between September 2017 and December 2021 were included in the study. Clinical and treatment data were obtained from medical records. Patient outcomes were assessed in-hospital, at a 1- and 12-month follow-up. (3) Results: There were 235 PERT activations. The risk of early mortality was low in 51 patients (21.8%), intermediate–low in 83 (35.3%), intermediate–high in 80 (34.0%) and high in 21 (8.9%) patients. Anticoagulation alone was the most frequently administered treatment in all patient subgroups (altogether 84.7%). Systemic thrombolysis (47.6%) and interventional therapy (52%) were the prevailing treatment options in high-risk patients. The in-hospital mortality was 6.4%. The adverse events during 1-year follow-up included five deaths, two recurrent VTE and two minor bleeding events. (4) Conclusions: Our initial 5-year experience showed that the activity of the local PERT facilitated patient-tailored decision making and the access to advanced therapies, with subsequent low overall mortality and treatment complication rates, confirming the benefits of PERT implementation.

## 1. Introduction

Pulmonary embolism (PE) is the third most frequent acute cardiovascular condition worldwide [1]. The incidence is 100–200 per 100,000 inhabitants each year [2], generating annual costs ranging to EUR 8.5 billion in the European Union alone [3]. The major burden of PE to the public health implies the need to optimize the strategies of PE diagnosis and management.

Given the diversity of PE clinical manifestation and multiple therapeutic interventions available in the acute PE [3], implementation of the optimal patient-tailored treatment is of the utmost importance. The choice of the optimal therapy should take into account the risk of early mortality and the risk of treatment-associated complications [4]. Generally, systemic thrombolytic therapy is recommended for patients with high-risk PE. However, the rate of major bleeding during systemic thrombolysis ranges to 20%, with the rate of intracranial bleeding up to 3% [5,6]. In addition, there are numerous patients in whom thrombolysis is initially contraindicated or has failed. In such patients, surgical or percutaneous catheter-directed therapy are viable alternative treatment options.

Regarding the complex qualification of PE patients for interventional treatment and the delicate balance between the risk of death due to the disease itself and the risk of treatment-associated complications [6], the concept of multidisciplinary PE response teams (PERTs) emerged in 2012 at the Massachusetts General Hospital [5]. By gathering experts from various disciplines, including interventional cardiology, cardiothoracic surgery, emergency medicine and intensive care within a rapid real-time consultation, the aim of PERTs is to optimize and accelerate treatment implementation in PE patients at intermediate–high and high risk of mortality [5,7]. 

The first results indicate that the implementation of PERTs improved the efficiency of treatment initiation and decreased both the hospital length of stay and the generated costs [6,7], although the clear mortality benefit remains to be demonstrated [6,7,8]. Following the European Society of Cardiology (ESC) recommendation to set up the local interdisciplinary PERTs for PE management [1], the Centre for the Management of Pulmonary Embolism (CELZAT) in Warsaw was established in 2017. The main goal of CELZAT is to improve patient prognosis by developing a model of interdisciplinary, comprehensive care for patients with PE, with particular focus on the population of patients with contraindications to standard pharmacological treatment, who require complex qualification for the interventional treatment [9]. Here, we report on the characteristics and outcomes of patients consulted by CELZAT.

## 2. Materials and Methods

### 2.1. Algorithm of CELZAT Activation

CELZAT was created by experts from the Department of Pulmonary Circulation, Thromboembolic Diseases and Cardiology, European Health Center in Otwock, Poland; 1st Chair and Department of Cardiology, Medical University of Warsaw, Poland; and Department of Cardiac Surgery, Medicover Hospital, Warsaw, Poland. The CELZAT project has been implemented in collaboration with Professor Richard Channik from the Massachusetts General Hospital in Boston, the creator of the world’s first interdisciplinary model of care for patients with pulmonary embolism, who acts as the Honorary Consultant.

An algorithm of the CELZAT activation consists of four stages (Figure 1). In the first stage, in patients with suspected acute PE, a thorough clinical assessment and risk stratification according to Pulmonary Embolism Severity Index (PESI) and simplified PESI (sPESI) is conducted by the treating physician. Based on the clinical picture, laboratory parameters and imaging finding, patients at high or intermediate–high-risk of mortality are identified. 

The diagnosis of high- and intermediate–high-risk of mortality PE is followed by CELZAT activation via a phone call to an emergency number, operating 24 h per day, 7 days per week. Subsequently, an interdisciplinary teleconsultation is performed within 30 min, including the treating physician, interventional cardiologist, clinical radiologist, intensive care specialist, anesthesiologist and cardiothoracic surgeon. Depending on the patient’s clinical condition and comorbidities, the expert panel may be extended by other specialists, such as a neurologist, general surgeon or vascular surgeon. 

The analysis of subsequent patients is conducted with the use of an online teleconsultation platform (Invisium MED, Ives-System, Warsaw, Poland). After logging in to the platform via a standard web browser, the analysis of clinical data and the results of additional examinations, which had previously been placed on the virtual drive, is performed during a real-time audiovisual consultation. The decisions made during the teleconsultation include (i) a possibility of pharmacological treatment optimization, (ii) indications for respiratory therapy, (iii) the use of extracorporeal membrane oxygenation (ECMO) and other forms of extra corporeal life support, (iv) indications for further invasive diagnostics (selective pulmonary angiography) and eventually percutaneous treatment and (v) indications for surgical treatment.

After the intervention, the patients are hospitalized in the Intensive Care Unit or the Cardiac Intensive Care Unit to stabilize the general condition, normalize hemodynamic parameters and monitor and treat the possible complications of the therapy. Particular emphasis is placed on the potential mechanical complications related to the percutaneous therapy, such as pulmonary dissection or perforation, cardiac tamponade and vascular access complications, as well as systemic complications, including contrast-induced acute kidney injury, arrhythmias, hypotension, hemolysis or bleeding.

### 2.2. Patient Enrollment and Data Collection

All patients diagnosed with PE who presented to any of the participating centers between September 2017 and December 2021 were included in the study. Information about clinical and treatment data was obtained from medical records, including (i) demographic data; (ii) symptoms and signs at presentation; (iii) risk factors of venous thromboembolism (VTE); (iv) comorbidities; (v) relevant laboratory and imaging findings (concentrations of cardiac troponins and natriuretic factors, features of RV overload on echocardiogram or computed tomography); (vi) VTE location, including the presence of deep vein thrombosis; (vii) in-hospital pharmacotherapy and interventional therapy; (vii) the need for endotracheal intubation, ECMO and admission to intensive care unit; and (viii) in-hospital and 1-month outcomes (mortality, recurrent PE or DVT and bleeding complications, as defined by the International Society of Thrombosis and Hemostasis, ISTH).

### 2.3. Assessment of PE Severity 

The severity of PE was each time categorized into high, intermediate–high, intermediate–low or low, according to the most recent ESC guidelines [1]. In all patients, the PESI and sPESI were calculated. High-risk PE was defined as confirmed acute PE with hemodynamic instability, i.e., clinical symptoms of cardiogenic shock or persistent hypotension (systolic blood pressure (BP < 90 mmHg or systolic BP drop ≥ 40 mmHg, lasting longer than 15 min and not caused by new-onset arrhythmia, hypovolemia or sepsis). The intermediate–high-risk group included patients who were hemodynamically stable but had features of RV overload (dysfunction on echocardiography or dilation on computed tomography pulmonary angiogram, CTPA) and laboratory marker of myocardial damage (cardiac troponins level above the institution-specific cut-off values). Intermediate–low-risk was defined as the presence of RV overload on echocardiography or CTPA, or elevated level of troponins, or PESI class III or higher, or at least 1 point in sPESI. The low-risk category involved patients in the PESI class I or II, or 0 points in sPESI. 

### 2.4. Treatment and Outcomes

Therapeutic interventions in hospital were recorded for each patient and involved: anticoagulation alone, systemic thrombolysis or interventional treatment. Anticoagulation was defined as the administration of the following: unfractionated heparin (UFH), low molecular weight heparin (LMWH), vitamin K antagonists (VKA) or direct oral anticoagulants (DOACs) without any additional therapies. Systemic thrombolysis referred to the intravenous administration of recombinant tissue plasminogen activator (rtPA). Catheter-directed procedures included catheter-directed thrombectomy (CDT), catheter-directed thrombolysis (CDL) and surgical embolectomy.

Interventional treatment was applied to patients with cardiogenic shock or significant hemodynamic instability who either were non-responsive or had contraindications to standard thrombolytic therapy. Catheter-directed thrombectomy or thrombolysis were preferred in patients at high perioperative risk of mortality and those who were disqualified from pulmonary embolectomy due to logistical reasons (lack of technical possibilities to transport the patient to the embolectomy-performing center, for example due to hemodynamic instability). Catheter-directed thrombectomy (CDT) was performed using the Angiojet^TM^ Rheolytic Thrombectomy System (Boston Scientific, Marlborough, MA, USA), Cleaner XTTM Rotational Thrombectomy System (Argon Medical Devices, Athens, TX, USA), or Indigo CAT8 XTORQ system (Penumbra, Alameda, CA, USA), depending on the anatomical conditions and morphology of thromboembolic lesions. The rate of catheter-directed thrombolysis (CDL), surgical embolectomy, ECMO or inferior vena cava (IVC) filter placement was also recorded.

Patient outcomes were assessed in-hospital and at 1- and 12-month follow-ups. Follow-ups included (i) mortality, (ii) stroke, (iii) recurrent PE/DVT and (iv) bleeding complications as defined by the ISTH. 

### 2.5. Statistical Analysis

Statistical analysis was conducted using IBM SPSS Statistics, version 27.0 (IBM, Sheffield, UK). Categorical variables were presented as number and percent. Continuous variables were presented as mean and standard deviation or median with interquartile range, depending on the distribution. A *p*-value below 0.05 was considered significant.

## 3. Results

### 3.1. Baseline Characteristics

During the 52-month enrollment period, there were 235 CELZAT activations: 104 in Medical University of Warsaw (44.3%), 116 in European Health Centre Otwock (49.3%) and 15 Medicover Hospital (6.4%). Patients’ characteristics at admission are presented in Table 1. The mean age was 60.3 ± 16.8 years, and the majority of patients were men (53.6%). The most common symptom at admission was dyspnea at minimal exertion (New York Heart Association [NYHA] functional class III; 42.0%) or at rest (NYHA class IV; 33.3% patients). Other symptoms included chest pain, syncope, cough and pneumonia, which were present in 31.9%, 16.6%, 15.7% and 13.2% of patients, respectively. The least common symptom was hemoptysis in only 5.5% of cases. Malignancy was the most frequent PE risk factor (34.0%). Besides malignancy, obesity (27.2%) and recent hospitalization (25.5%) were the most common, followed by smoking (24.7%). Thirty-nine patients had a history of previous DVT (16.6%), and 13 patients had previous PE (5.5%). Previous COVID-19 infection was risk factor for PE in 14 cases (5.9%).

### 3.2. Characteristics of Pulmonary Embolism

The risk of early mortality was low in 51 patients (21.8%), intermediate–low in 83 (35.3%), intermediate–high in 80 (34.0%) and high in 21 (8.9%) patients (Figure 2). The vast majority of patients had thrombus located bilaterally (77.4%) and centrally (82.6%). The central location of the thrombus was defined as the saddle, main pulmonary artery, lobar artery and intracardiac location. Peripheral location was defined as the segmental and subsegmental artery. Patients with high-risk PE presented more often with bilateral PE (95.2%) and central PE (100.0%), compared to other risk categories (Figure 3 and Figure 4). However, bilateral and central PE was also the most frequent phenotype in all other subgroups of patients.

The characteristics of PE categorized according to the risk of early mortality are showed in Table 2. The most common parameter of PE severity was RV overload on echocardiography or CTPA (67.2%), followed by elevated concentration of natriuretic peptides (73.6%) and troponins (55.7%). PE was accompanied by DVT in 51.1% of all patients. Twenty-six patients required endotracheal intubated (11.1%), fifteen high-risk patients required ECMO (6.4%) and eighty-four patients from intermediate–high and high-risk of mortality subgroups (83.2% of both subgroups) were admitted to the ICU. One hundred twenty-six patients were in the PESI class III or higher (53.6%). A high-risk score (at least 1 point on the sPESI scale) was present in 162 patients (68.9%). 

### 3.3. Treatment

The details of in-hospital and post-discharge treatment according to mortality risk groups are presented in Table 3. Anticoagulation alone was the most frequently administered treatment, received by 84.7% of patients and this trend applied to all risk subgroups, except for high-risk patients, where systemic thrombolysis (47.6%) and interventional therapy (52%; CDT/CDL 28.6% and surgical embolectomy 23.4%) were the prevailing treatment options. 

Systemic thrombolysis was administered in one intermediate–low-risk patient (1.2%) and four intermediate–high-risk patients (5.0%). Eleven patients (9.4%) were treated with catheter-directed procedures (seven patients with catheter-directed thrombectomy and four with catheter-directed thrombolysis. Fifteen patients (6.4%) with intermediate–high or high-risk PE underwent surgical embolectomy. Nineteen patients (8.1%) received IVC filter. The high rate of IVC use was due to the characteristics of the patients consulted by CELZAT. In addition to PE, the consulted patients had comorbidities that are contraindications to standard anticoagulant therapy. The most common indications for IVC filter were increased risk of bleeding in the context of malignancy and status after orthopedic surgery for massive trauma. In a few cases, the indication for IVC filter implantation was the recurrence of PE during standard anticoagulation treatment, which occurred most frequently in patients with oncological metastasis.

Combined therapy was performed in 10 patients. Four patients were treated by catheter-directed thrombectomy with subsequent transcatheter thrombolysis. Three of them received systemic thrombolysis followed by surgical embolectomy. In one case the transcatheter procedure was associated with further systemic thrombolysis and in one case the transcatheter procedure was followed by surgical embolectomy.

At discharge, the majority of all patients received DOACs (55.0%), followed by LMWH (35.0%) and VKA (10.0%). Among patients who received LMWH, 46 had a coexistent malignancy.

### 3.4. Outcomes

The in-hospital, 1-month follow-up and 12-month follow-up outcome events according to mortality risk groups are showed in Table 4. The rate of in-hospital mortality was 6.4% (15/235 patients: 8 in the high-risk subgroup, 4 in the intermediate–high-risk subgroup and 3 in the intermediate–low-risk subgroup). Three of them suffered from malignancy. All patients presented with dyspnea NYHA class IV (thirteen patients) or III (two patients). Thirteen patients were admitted to ICU and eight required endotracheal intubation. Nine patients received only anticoagulation, in two patients systemic thrombolysis was implemented, two patients underwent CDT and another two received combined therapy. 

There were twelve minor bleeding events (5.1%) and four of them required reduced dosage of anticoagulant therapy. There were seven major bleeding events (3.0%), which required blood transfusion and the modification of anticoagulation treatment. There were three strokes (1.3%): one in the intermediate–low-risk patient treated with anticoagulation only and two in high-risk patients treated with thrombolysis. There was one fatal recurrence of PE in the intermediate–high patient treated with systemic full-dose thrombolysis and one recurrence of DVT, associated with major bleeding and the modification of anticoagulant treatment. In the last case, an IVC filter was applied to protect patients during the next weeks. 

At 1-month follow-up, there was one death and one stroke. Two bleeding events were registered. One of them was minor and did not require future action. Another one was major and required medical attention. The latter patient died during the 1-year follow-up. Other adverse events during the 1-year follow-up included five deaths, one recurrent PE, one recurrent DVT and two minor bleeding events.

## 4. Discussion

The goal of PERT is to deliver rapid, interdisciplinary care to patients with PE to facilitate the access to advanced treatment and improve outcomes. Since the launch of the first PERT in 2012, the idea of PERT has spread worldwide. The results of hitherto published observational studies suggest that implementation of PERT increased patients’ access to advanced therapies (systemic thrombolysis, catheter-directed procedures, surgery) without increasing the number of bleeding complications [6,7,10]. Moreover, recent studies comparing outcomes in the pre- and post-PERT era showed that the availability of multidisciplinary PERT was associated with decreased 30-day mortality, especially among high-risk patients, without incurring additional hospital costs or protracting hospital length-of-stay [11,12]. However, the clear benefits of PERT in terms of mortality have not been confirmed in all studies [6], which is likely due to the limited sample size and short follow-up time of patients included in these studies [6,7,8,10,11,12,13,14,15,16,17,18]. Hence, there is an unmet need to collect more data regarding the performance of PERT and share them with the medical community.

The current study is the first report concerning the activity of the local PERT, CELZAT. CELZAT operates in accordance with the standards outlined in the position paper of Polish PERT Initiative [19], with the primary objective to deliver care to intermediate–high and high-risk PE patients. However, in our cohort, less than half of patients presented with intermediate–high-risk PE (34.0%) and high-risk PE (8.9%). We registered numerous activations of PERT in the intermediate–low (35.4%) and low-risk (21.7%) subgroups, which might be due to two reasons. First, substantial efforts have been made to raise the awareness of the local PERT via journal publications [19], conference presentations and social media, which prompted the treating physicians to contact PERT experts in case of any acute PE. Second, the majority of intermediate–low and low-risk patients in our cohort presented with bilateral and central PE, raising concerns regarding the potential risk for sudden clinical deterioration and death despite normal hemodynamics at initial assessment. PERT activation in low-risk patients has been noticed in other papers reporting PERT activity [8,14], introducing the concept of a “high-risk patient with a low-risk PE” [8]. Further research is required to investigate the clinical benefits and economic efficacy of PERT activation in low-risk patients. For example, PERT activation might be rational in order to optimally manage the low-risk patients with contraindications to anticoagulation or with comorbidities requiring a multidisciplinary approach. 

Anticoagulation was the most frequently administered treatment in all risk subgroups except for high-risk patients (approximately 85% of our cohort), in accordance with previous reports [8,14]. The most frequently administered anticoagulants at discharge were DOACs (55.0%), whereas only a minority of patients received VKA (10.0%). LMWH was received by a substantial number of patients at discharge (35.0%), the majority of whom presented with a co-existing malignancy. The choice of anticoagulation for cancer-associated VTE is a major therapeutic challenge due to a delicate balance between the recurrent thromboembolic and bleeding events in these patients [20]. LMWH has traditionally been the standard treatment for cancer-associated VTE due to higher efficacy and comparable safety, compared to VKA [21]. However, the results of recent trials comparing LMWH to DOAC showed that DOAC might be more effective than LMWH at preventing recurrent VTE in cancer patients but at the cost of increased bleeding, especially in patients with gastrointestinal (GI) and genitourinary (GU) tract cancer. Consequently, the ISTH International Initiative on Thrombosis and Cancer (ITAC) guidelines state that DOACs can be used as first line treatment for cancer-associated thrombosis in non-GI/GU cancer patients at low bleeding risk. Among patients with GI/GU cancer-associated thrombosis, LMWH is still preferred [22]. Accordingly, the ESC guidelines recommend that edoxaban or rivaroxaban should be considered as an alternative to LMWH, with special caution for patients with GI cancer [1]. However, as demonstrated in our cohort, clinicians are still reluctant to prescribe DOAC in cancer patients and more evidence-based data are required to establish the optimal treatment regimen in this challenging population.

The proportion of patients receiving any advanced therapy (systemic thrombolysis, surgical embolectomy or catheter-directed procedures) in our study was 15.3% (81.0% of patients in the high-risk subgroup and 21.3% in the intermediate–high-risk group), confirming the widespread access of patients to the advanced treatment methods within PERT. As showed by the National PERT Consortium™ multicenter registry, this proportion varies between institutions, ranging from 16% to 46%, underlying the need to share the institutional PERT experiences [8,17]. In the entire cohort, systemic thrombolysis and surgical embolectomy were the most common advanced therapies (6.4% each group), followed by catheter-directed procedures (4.7%).

Although the evidence-based data regarding the effect of catheter-directed procedures on mortality are still pending, catheter-directed therapies has become an important and less-invasive treatment option both in intermediate–high and high-risk patients, either to temporarily stabilize the patient before surgical embolectomy or as a final therapy if hemodynamical stability is restored [23]. Although catheter-directed procedures are dedicated to intermediate–high and high-risk patients, they have also been applied to one intermediate–low-risk patient, who deteriorated hemodynamically following the initial assessment. Among eleven patients treated with catheter-directed procedures, there were four in-hospital deaths (one due to a stroke) and two major bleeding events. Although the efficacy and safety of catheter-directed procedures has been shown in other PERT reports, our data indicate the need for cautious patient qualification for these therapies, as they may be associated with complications [10,14,15,16,17,18].

The in-hospital mortality rate in our cohort was 6.4%, which is slightly lower than reported by other studies (8 and 14%) [10,14,15,16,17,18], likely due to the large percentage of low and intermediate–low-risk patients in our cohort. We registered only one death during the 1-month follow-up period, indicating the efficacy of rapid treatment implementation by PERT. The mortality rate in high-risk patients was high (38.1%), as compared to relatively low mortality in other subgroups (0.0% in low-risk, 3.6% in intermediate–low-risk and 5.0% in intermediate–high-risk). Although the mortality rate in high-risk PE patients remains unacceptably high, recent trend analyses showed a substantial decrease in mortality due to high-risk PE from 72.7% in 1999 to 49.8% in 2017 [24,25]. In high-risk PE, systemic thrombolysis is the first-line treatment and surgical embolectomy is recommended when systemic thrombolysis is contraindicated or has failed [1]. High-risk patients constituted approximately 9% of our cohort and were all treated with either systemic thrombolysis or, in the presence of contraindications, interventional treatment (surgical embolectomy or catheter-directed procedures). The detailed management of some of these patients has been previously published [26,27]. The high risk of mortality in high-risk PE patients, despite multidisciplinary PERT management, underlines the need for further treatment optimalisation in this challenging population.

Any bleeding occurred in 8.1% and major bleeding in 3.0% of patients, which is lower than the previously reported bleeding rates, ranging from 11–13% for any bleeding and 4–13% for major events [10,13,15,16,17,18]. In our study, in-hospital bleeding events occurred in all risk groups, and five of the bleeding events occurred following systemic thrombolysis. These findings remain consistent with previous reports [8]. There were two bleeding episodes during the 1-month follow-up. However, these findings should be interpreted with caution due to a very low number of incidents.

Finally, we observed PERT activation cases in patients with non-thromboembolic embolism, such as those with iatrogenic PE [28,29]. In these situations, individualized risk stratification by PERT allowed the determination as to whether an interventional or conservative approach is more beneficial. Hence, although the primary goal of PERT is to consult patients with thromboembolism, the availability and expertise of PERT provided additional clinical benefits in non-thromboembolic, difficult clinical scenarios.

## 5. Limitations

Our analysis had several limitations. First, over 50% our cohort consisted of low- and intermediate–low-risk patients. Hence, the reported results are preliminary and should be confirmed in a larger cohort and in higher risk patients. The large proportion of low- and intermediate–low-risk patients does not allow an effective assessment of treatment complications and overall mortality. Further studies should specifically focus on patients in the intermediate–high- and high-risk groups. Second, we report on the activity of a single PERT, created at highly specialized academic medical centers with access to interdisciplinary care. Although different institutional PERTs have similar operating models, the local factors may affect the individual therapeutic choices, as reflected by different rates of advanced therapies, depending on the center [10,14,15,16,17,18]. Therefore, our results cannot be directly extrapolated to other institutions. Third, the efficacy and safety outcomes are limited to 12-month observation period, which does not allow the deriving of conclusions regarding the long-term benefits of PERT implementation. Since the goal of this study was to report on the characteristics and outcomes of patients consulted by the local PERT, CELZAT, our study design precluded comparison between some subgroups. It would be useful to compare patient outcomes before and after CELZAT implementation. Unfortunately, the lack of standardized reporting of patients with PE before CELZAT implementation made this analysis not feasible. In addition, subanalyses of outcomes in PE subgroups depend on the initial risk of mortality and/or comorbidities, such as malignancy. Fourth, because of the lack of data regarding the family history of VTE, we could not take this risk factor into account, which may be an important cause of VTE, especially in young patients. Finally, we did not evaluate the cost-efficacy of PERT implementation (length of hospital stay, costs of PERT activation, patient’s quality of life following the PE episode). Hence, we cannot derive conclusions regarding the economic effects of PERT implementation in our institution.

## 6. Conclusions

We provide the initial experience regarding the 5-year activity of the local PERT (CELZAT). The implementation of a multidisciplinary PERT enabled patient-tailored decision making and facilitated the access to advanced therapies, with subsequent low overall mortality and treatment complication rates. Our findings add to the previously described experiences derived from other institutional PERTs, confirming the benefits of PERT implementation. There is a need for multicenter collaboration between the local PERTs to derive firm conclusions regarding the favorable effect of PERT activity on patient outcomes.

## Figures and Tables

**Figure 1 jcm-11-03812-f001:**
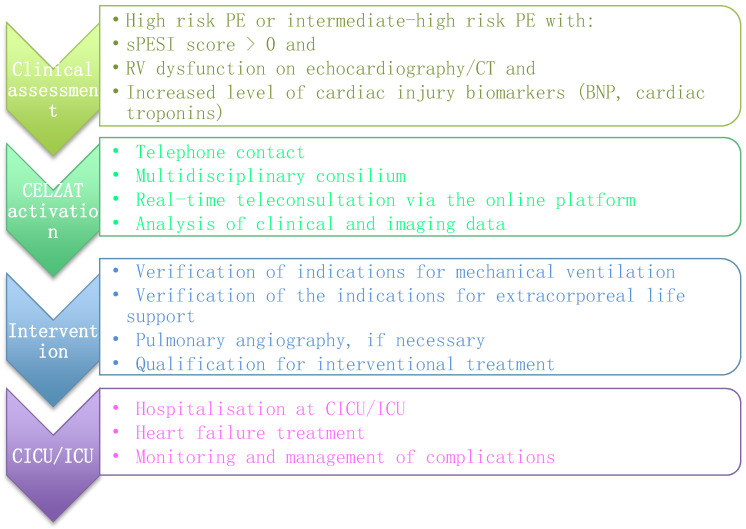
Activation flowchart of CELZAT. PE—pulmonary embolism, sPESI—simplified Pulmonary Embolism Severity Index, RV—right ventricle, CT—computed tomography, BNP—brain natriuretic peptide, CICU—cardiac intensive care unit, ICU—intensive care unit.

**Figure 2 jcm-11-03812-f002:**
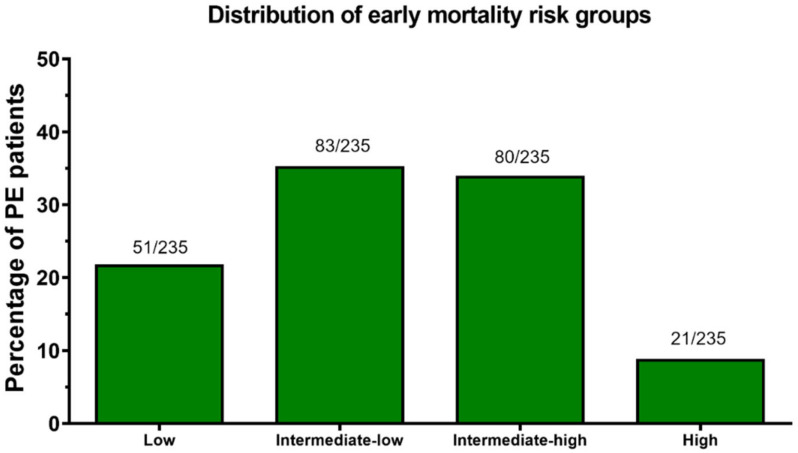
Distribution of the early mortality risk groups in the study population.

**Figure 3 jcm-11-03812-f003:**
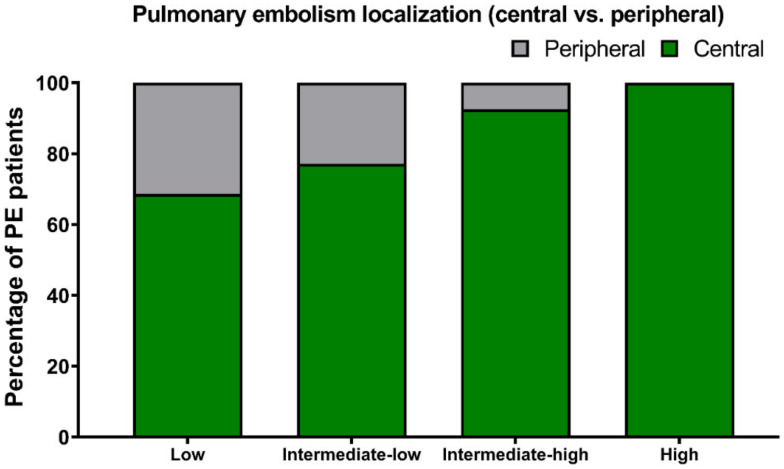
Pulmonary embolism localization (central vs. peripheral) according to the risk of early mortality.

**Figure 4 jcm-11-03812-f004:**
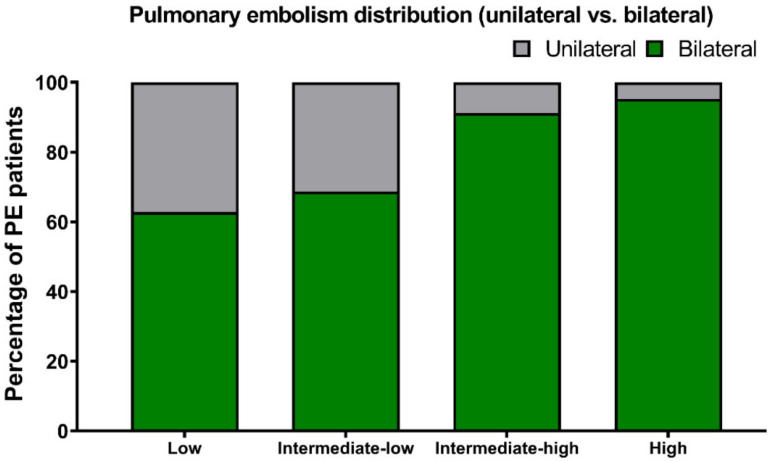
Pulmonary embolism distribution (unilateral vs. bilateral) according to the risk of early mortality.

**Table 1 jcm-11-03812-t001:** Baseline characteristics of PE patients.

	Patients (n = 235)
Baseline characteristics	
Age, years (mean ± SD)	60.3 ± 16.8
Sex, male (n, %)	126 (53.6%)
Symptoms on admission	
Dyspnea (NYHA; n, %)	
I-II	43 (24.7%)
III	73 (42.0%)
IV	58 (33.3%)
Chest pain (n, %)	75 (31.9%)
Syncope (n, %)	39 (16.6%)
Cough (n, %)	37 (15.7%)
Pneumonia (n, %)	31 (13.2%)
Hemoptysis (n, %)	13 (5.5%)
Comorbidities (n, %)	
Coronary artery disease	24 (10.2%)
Congestive heart failure	22 (9.4%)
Atrial fibrillation	16 (6.8%)
Arterial hypertension	113 (48.1%)
COPD	11 (4.7%)
Diabetes mellitus	41 (17.4%)
Obesity	64 (27.2%)
Chronic kidney disease	17 (7.2%)
Stroke	15 (6.4%)
Depression	12 (5.1%)
Malignancy	80 (34.0%)
Thrombophilia	12 (5.1%)
Other VTE risk factors (n, %)	
Smoking	58 (24.7%)
Indwelling catheter	7 (3.0%)
Hormonal therapy	13 (5.5%)
Reduced mobility	27 (11.5%)
Recent hospitalization	60 (25.5%)
Recent surgery	28 (11.9%)
Recent trauma	14 (6.0%)
Prior PE	13 (5.5%)
Prior DVT	39 (16.6%)
COVID-19 infection	14 (6.0%)

NYHA—New York Heart Association, COPD—chronic obstructive pulmonary disease, VTE—venous thromboembolism, DVT—deep vein thrombosis.

**Table 2 jcm-11-03812-t002:** Characteristics of PE categorized according to the risk of early mortality.

PE Risk Category	Low (n = 51)		Intermediate–Low (n = 83)		Intermediate–High (n = 80)		High (n = 21)		All (n = 235)	
PE location (n, %)										
Bilateral	32	62.7%	57	68.7%	73	91.3%	20	95.2%	182	77.4%
Unilateral	19	37.3%	26	31.3%	7	8.7%	1	4.8%	53	22.6%
Central	35	68.6%	64	77.1%	74	92.5%	21	100.0%	194	82.6%
Peripheral	16	31.4%	19	22.9%	6	7.5%	0	0.0%	41	17.4%
Saddle	1	2.0%	8	9.6%	26	32.5%	9	42.9%	44	18.7%
Main pulmonary artery	19	37.3%	28	33.7%	61	76.3%	16	76.2%	124	52.8%
Lobar artery	33	64.7%	61	73.5%	63	78.8%	19	90.5%	176	74.9%
Segmental artery	43	84.3%	70	84.3%	51	63.8%	14	66.7%	178	75.7%
Intracardiac	0	0.0%	0	0.0%	7	8.8%	2	9.5%	9	3.8%
Parameters of PE severity (n, %)										
RV dysfunction (ECHO)	0	0.0%	21	25.3%	70	87.5%	16	76.2%	107	45.5%
RV dilation (CTPA)	0	0.0%	10	12.0%	31	38.8%	10	47.6%	51	21.7%
↑ Troponin	0	0.0%	38	45.8%	80	100.0%	14	66.7%	131	55.7%
↑ Natriuretic peptides	20	39.2%	61	73.5%	74	92.5%	18	85.7%	173	73.6%
DVT	20	39.2%	34	41.0%	54	67.5%	12	57.1%	120	51.1%
PESI class (n, %)										
I–II	36	70.6%	38	45.8%	34	42.5%	1	4.8%	109	46.4%
III	7	13.8%	15	18.1%	28	35.0%	1	4.8%	51	21.7%
IV	4	7.8%	19	22.9%	10	12.5%	5	23.8%	38	16.2%
V	4	7.8%	11	13.2%	8	10.0%	14	66.6%	37	15.7%
Score (median, IQR)	65 (49–88)		93 (74–113)		89 (71–105)		144 (123–195)		88 (69–114)	
sPESI (n, %)										
Low risk	32	62.7%	21	25.3%	20	25.0%	0	0.0%	73	31.1%
High risk	19	37.3%	62	74.7%	60	75.0%	21	100.0%	162	68.9%
Clinical severity (n, %)										
Intubation	0	0.0%	1	1.2%	13	16.3%	12	57.1%	26	11.1%
ECMO support	0	0.0%	0	0.0%	10	12.5%	5	23.8%	15	6.4%
ICU admission	24	47.1%	37	44.6%	63	78.8%	21	100.0%	145	61.7%

RV—right ventricle, ECHO—echocardiography, CTPA—computed tomography pulmonary angiogram, ECMO—extracorporeal membrane oxygenation.

**Table 3 jcm-11-03812-t003:** In-hospital and post-discharge treatment according to mortality risk groups.

PE Risk Category	Low (n = 51)		Intermediate–Low (n = 83)		Intermediate–High (n = 80)		High (n = 21)		All (n = 235)	
In-hospital (n, %) *										
Anticoagulation alone	51	100.0%	81	97.6%	63	78.8%	4	19.0%	199	84.7%
Systemic thrombolysis	0	0.0%	1	1.2%	4	5.0%	10	47.6%	15	6.4%
CDT/CDL	0	0.0%	1	1.2%	2	5.0%	6	28.6%	11	4.7%
Surgical embolectomy	0	0.0%	0	0.0%	10	12.5%	5	23.4%	15	6.4%
IVC filter	1	2.0%	8	9.6%	7	8.8%	3	14.3%	19	8.1%
At discharge (n, %) **	n = 51		n = 80		n = 76		n = 13		n = 220	
VKA	3	5.9%	5	6.3%	12	15.8%	2	15.4%	22	10.0%
DOAC	40	78.4%	41	51.2%	37	48.7%	3	23.1%	121	55.0%
LMWH	8	15.7%	34	42.5%	27	35.5%	8	61.5%	77	35.0%

* The number of patients might exceed 235 due to combined therapies applied to some patients (e.g., interventional therapy on top of anticoagulation or systemic thrombolysis). ** The number of patients at discharge is affected by in-hospital mortality. CDT—catheter directed thrombectomy, CDL—catheter directed thrombolysis, IVC—inferior vena cava, VKA—vitamin K antagonists, DOAC—direct oral anticoagulants, LMWH—low molecular weight heparin.

**Table 4 jcm-11-03812-t004:** In-hospital, 1-month and 12-month follow-up outcome events according to the mortality risk groups.

PE Risk Category	Low (n = 51)		Intermediate–Low (n = 83)		Intermediate–High (n = 80)		High (n = 21)		All (n = 235)	
In-hospital events (n, %)										
Death	0	0.0%	3	3.6%	4	5.0%	8	38.1%	15	6.4%
Stroke	0	0.0%	1	1.2%	0	0.0%	2	9.5%	3	1.3%
Major bleeding	0	0.0%	4	4.8%	1	1.3%	2	9.5%	7	3.0%
Minor bleeding	1	2.0%	7	8.4%	2	2.5%	2	9.5%	12	5.1%
Recurrent PE	0	0.0%	0	0.0%	1	1.3%	0	0.0%	1	0.4%
Recurrent DVT	0	0.0%	1	1.2%	0	0.0%	0	0.0%	1	0.4%
1-month follow-up **										
Death	1	2.0%	0	0.0%	0	0.0%	0	0.0%	1	0.4%
Stroke	1	2.0%	0	0.0%	0	0.0%	0	0.0%	1	0.4%
Major bleeding	1	2.0%	0	0.0%	0	0.0%	0	0.0%	1	0.4%
Minor bleeding	0	0.0%	1	1.2%	0	0.0%	0	0.0%	1	0.4%
Recurrent PE	0	0.0%	0	0.0%	0	0.0%	0	0.0%	0	0.0%
Recurrent DVT	0	0.0%	0	0.0%	0	0.0%	0	0.0%	0	0.0%
12-month follow-up **										
Death	1	2.0%	1	1.2%	2	2.6%	2	15.4%	6	2.7%
Stroke	0	0.0%	0	0.0%	0	0.0%	0	0.0%	0	0.0%
Major bleeding	0	0.0%	0	0.0%	0	0.0%	0	0.0%	0	0.0%
Minor bleeding	0	0.0%	1	1.2%	0	0.0%	1	7.7%	2	0.9%
Recurrent PE	1	2.0%	0	0.0%	0	0.0%	0	0.0%	1	0.4%
Recurrent DVT	0	0.0%	1	1.2%	0	0.0%	0	0.0%	1	0.4%

** The number of patients at follow-up is affected by in-hospital and follow-up mortality.

## Data Availability

Raw data are available upon request to the corresponding author.

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
