# Peer review of "Characteristics and Outcomes of Patients Consulted by a Multidisciplinary Pulmonary Embolism Response Team: 5-Year Experience"

_jcm, 2022, doi:10.3390/jcm11133812_

Round 1

Reviewer 1 Report

Pietrasik et al. studied the characteristics and outcome of patients consulted by a multidisciplinary pulmonary embolism response team. To the best of my knowledge, PE response team is rare in the world. Your initial 5-year experience contributed to be low overall mortality and treatment complication rates. This study is interested; however, I have some questions.

  1. If high or intermediate-high risk of PE was detected, physician consult CELZAT by phone in 24 hours per day, 7 days per week. Subsequently, interdisciplinary teleconsultation is performed within 30 minutes. I think that this consultation system is very hard for team member. Do authors have any device for this system?
  1. Do authors have any data of PE mortality or PE-related complications before introduction of CELZET system and compare these data?
  1. I think that the rate of IVC filter use is slightly high. Authors should show the indication for IVC filter use.
  1. Do authors think what risk group have most benefit by CELZAR system?

Author Response

Pietrasik et al. studied the characteristics and outcome of patients consulted by a multidisciplinary pulmonary embolism response team. To the best of my knowledge, PE response team is rare in the world. Your initial 5-year experience contributed to be low overall mortality and treatment complication rates. This study is interested; however, I have some questions.

Together with my co-authors, I would like to thank the Reviewer for the review of the manuscript. Following the suggestions of the Reviewers, the changes have been made and marked in the current version of the manuscript. Detailed descriptions of the corrections and responses to the reviewers’ questions are provided below

  1. If high or intermediate-high risk of PE was detected, physician consult CELZAT by phone in 24 hours per day, 7 days per week. Subsequently, interdisciplinary teleconsultation is performed within 30 minutes. I think that this consultation system is very hard for team member. Do authors have any device for this system? 

Efficient response and 24/7 interdisciplinary consultation are possible due to the cooperation between our centers. Each center has a specialist on duty who can respond to a PERT activation at any time, similarly to the 24/7 interventional cardiology duty, which is present in Poland since the year 2004.

  1. Do authors have any data of PE mortality or PE-related complications before introduction of CELZET system and compare these data?

Unfortunately, we did not collect any data before CELZAT was launched and therefore we could not compare patients’ outcomes before and after the implementation of CELZAT. This is due to the lack of standardized reporting of PE patients and the incomplete medical records at our institutions before CELZAT implementation. We entirely agree with the Reviewer that such comparison would be very interesting and we added the lack of such analysis to the Limitations, as follows: “It would be useful to compare patient outcomes before and after CELZAT implementation. Lack of standardized reporting of patients with PE before CELZAT implementation made this analysis not feasible.”

  1. I think that the rate of IVC filter use is slightly high. Authors should show the indication for IVC filter use.

The high rate of IVC use is due to the characteristics of the patients consulted by CELZAT. In addition to pulmonary embolism, the consulted patients had comorbidities that are contraindications to standard anticoagulant therapy. The most common indications for IVC filter were an increased risk of bleeding in the context of malignancy and status after orthopedic surgery for massive trauma. In a few cases, the indication for IVC filter implantation was the recurrence of PE during standard anticoagulation treatment. This situation occurred most frequently in patients with oncological metastasis. We added this explanation to the Results.

  1. Do authors think what risk group have most benefit by CELZAT system?

In our opinion, intermediate- and high-risk patient groups benefit most from CELZAT. Access to advanced treatment techniques, such as percutaneous thrombectomy, transcatheter thrombolysis, surgical embolectomy, and inferior vena cava filter, provides personalized treatment to at-risk patients. Choosing the appropriate treatment technique requires expert knowledge, especially in patients with multiple comorbidities or contraindications to standard thrombolytic treatment, such as malignancies. This is made possible by the collaboration of various specialists through CELZAT.

Altogether, we thank the Reviewer for the detailed review of our paper and all the comments that allowed us to improve our manuscript.

Reviewer 2 Report

In the latest guidelines (ESC) a PE response team (PERT) have been suggested to facilitate treatment implementation in PE patients. However, the recommendation are weak due to lack of evidence.

 This is an interesting study reporting 5-year experience of PERT with regard to the characteristics and outcomes in patients diagnosed with PE between September 2017 and December 2021 in Poland. Clinical and treatment data was obtained from medical records. Patient outcomes were assessed in- hospital, at 1- and 12-month follow-up. In the present multi center study, there were 235 PERT activations between 2017 and 2021. The risk of early mortality was low or intermediate-low in over half of the patients, intermediate-high in 34.0% and high in 8.9% of the patients. Systemic thrombolysis and interventional therapy were the prevailing treatment options in high-risk patients. The in‑hospital mortality was 6.4%. The adverse events during 1-year follow-up included 5 deaths, 2 recurrent VTE and 2 minor bleeding events.

The authors conclude that there is a low overall mortality and treatment complication rates, confirming the benefits of PERT implementation.

This is an interesting study.  However, the study has limitations.

-          Most of the patients were in the low or intermediate low risk group and it is not possible to conclude that that the mortality rate or complication rates are low due to power in the study.

-          Was the design of the study retrospective?

-          It would have been interesting to have the total number of patients with PE admitted to the hospital in the same periode as a control group.

Minor comments; Brachets for the reference numbers are missing several times.

Author Response

In the latest guidelines (ESC) a PE response team (PERT) have been suggested to facilitate treatment implementation in PE patients. However, the recommendation are weak due to lack of evidence.

 This is an interesting study reporting 5-year experience of PERT with regard to the characteristics and outcomes in patients diagnosed with PE between September 2017 and December 2021 in Poland. Clinical and treatment data was obtained from medical records. Patient outcomes were assessed in- hospital, at 1- and 12-month follow-up. In the present multi center study, there were 235 PERT activations between 2017 and 2021. The risk of early mortality was low or intermediate-low in over half of the patients, intermediate-high in 34.0% and high in 8.9% of the patients. Systemic thrombolysis and interventional therapy were the prevailing treatment options in high-risk patients. The in‑hospital mortality was 6.4%. The adverse events during 1-year follow-up included 5 deaths, 2 recurrent VTE and 2 minor bleeding events.

The authors conclude that there is a low overall mortality and treatment complication rates, confirming the benefits of PERT implementation.

This is an interesting study.  However, the study has limitations.

We thank the reviewer for taking the time to evaluate our work. Thank you for appreciating our paper. As suggested by the Reviewers, changes have been made and highlighted in the current version of the manuscript. Detailed descriptions of the revisions and responses to the Reviewers' questions can be found below.

-          Most of the patients were in the low or intermediate low risk group and it is not possible to conclude that that the mortality rate or complication rates are low due to power in the study.

We are aware that the large proportion of low and intermediate-low risk patients is a limitation in assessing safety and mortality in the study population. As suggested by the Reviewer, we highlighted this in the Limitations as follows: "The large proportion of low- and intermediate-low-risk patients does not allow an effective assessment of treatment complications and overall mortality. It seems that further studies should focus on patients in the intermediate-high- and high-risk groups".

-          Was the design of the study retrospective?

The patients were prospectively consulted by PERT, but the analysis was entirely retrospective, based on the medical records collected between September 2017 and December 2021.

-          It would have been interesting to have the total number of patients with PE admitted to the hospital in the same periode as a control group.

Thank you for this interesting insight. In the current study, it was not possible to use patients admitted to the hospital during the same period as a control group. The reason for this was the lack of data collection on patients with PE but not consulted through CELZAT. In addition, intermediate-high and high-risk patients were regularly reported to CELZAT. Other patients remained in the lower risk groups which makes it possible that the two populations may differ.

Minor comments; Brachets for the reference numbers are missing several times.

Altogether, we are grateful for the in-depth revision of our manuscript and we hope that it will be considered for publication in “Journal of Clinical Medicine”. All the authors have approved the manuscript and agree with the corrections.

Round 2

Reviewer 1 Report

I am satisfied with authors’ responses.